# 🐢 TurtleBench: A Visual Programming Benchmark in Turtle Geometry

**Sina Rismanchian, Yasaman Razeghi, Sameer Singh, Shayan Doroudi**
University of California, Irvine
{srismanc,yrazeghi,sameer,doroudis}@uci.edu

## Abstract

While formal geometric reasoning may be difficult for humans without extensive training, humans seem to have the ability to intuitively reason about geometric patterns in images and scenes from a young age. In contrast, developing large multimodal models (LMMs) capable of similar feats represents a frontier in AI research. We introduce TurtleBench, a benchmark designed to evaluate LMMs' capacity to interpret geometric patterns—given visual examples, textual instructions, or both—and generate precise code outputs. Inspired by turtle geometry, a notion used to teach children foundational coding and geometric concepts, TurtleBench features tasks with patterned shapes that have underlying algorithmic logic. Unlike object detection tasks that typically do not involve understanding underlying patterns, this benchmark combines geometrical reasoning with image understanding. Our evaluation reveals that leading LMMs struggle significantly with these tasks, with GPT-4V achieving only 19% accuracy on the simplest tasks. TurtleBench highlights the gap between human and AI performance in intuitive and visual geometrical understanding, setting the stage for future research in this area.

## 1 Introduction

Geometric reasoning is a hallmark of human mathematical reasoning that has been studied since the Ancient Greeks. It was a task that attracted early artificial intelligence (AI) researchers and early efforts on building intelligent tutoring systems also focused on geometry. Yet much of the emphasis on geometric reasoning is on axiomatic-deductive geometry. Humans of all ages are naturally good at more intuitive kinds of geometric reasoning that inform how we see and navigate the world. One aspect of this is our ability to look at a geometric shape or complex pattern and construct an algorithm to generate that pattern. We believe this is a powerful task to evaluate large multimodal models (LMMs) for a number of reasons. First of all, constructing patterns in this way reflects an early programming paradigm for teaching kids programming, initially developed in the 1970s with the introduction of the Logo programming language (Papert, 1972, 1980). For several decades, children from a young age have been learning how to procedurally draw geometric patterns and other drawings using code in programming languages like Logo, Scratch, and Python—often as their first introduction to programming. Given LMMs' success in a variety of complex programming tasks, one might expect a programming task that children could solve to be easy. Second, recent research suggests that this ability to procedurally generate shapes may be more fundamental to our psychology than meets the eye. Spelke (2022) claims that from infancy (or even birth), humans have a set of six core knowledge systems, two of which contribute to our understanding of geometry: a *form* system and a *place* system. While the form system allows us to perceive the boundaries of objects, our core knowledge of places interprets geometry in terms of how to navigate an environment (Dillon, 2023). Taking this a step further, Sablé-Meyer et al. (2022) suggest that humans perceive shapes and patterns in terms of procedural programs that could generate them; they demonstrate that the time it takes

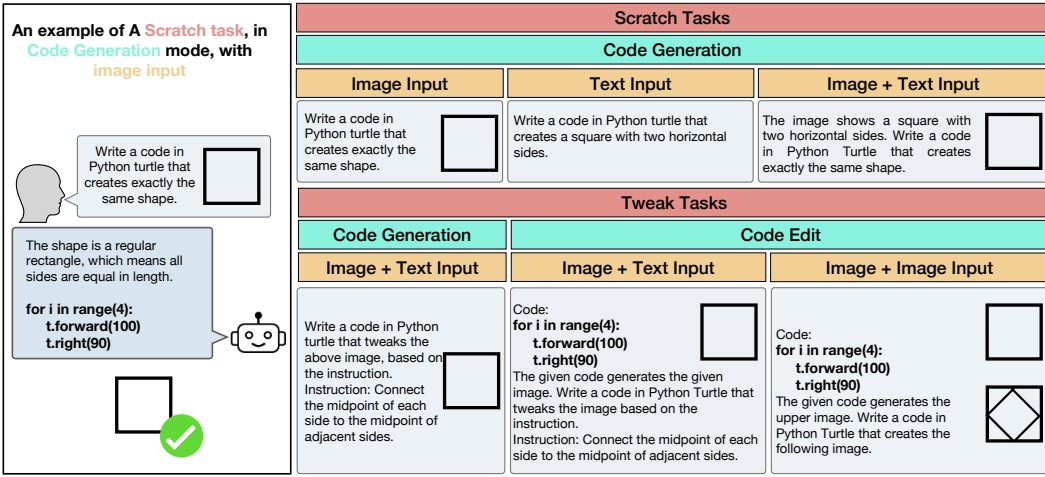

Figure 1: An illustration of existing types and modes in TurtleBench, A task may have a type of Scratch or Tweak, in a mode of code generation or code edit, with various modalities in the input.

for people to process these shapes correlates with the minimum description length of the shape in a Logo-like programming language.

In this work, we introduce TurtleBench, a set of manually crafted image/text to code tasks in turtle geometry (Papert, 1972; Abelson & diSessa, 1986) to evaluate the abilities of these models to combine visual pattern recognition, abstract geometrical reasoning, and Python programming. To ensure the visual inputs and the programming language remain straightforward, TurtleBench harnesses turtle geometry, a concept widely recognized for its effectiveness in introducing programming concepts to children within the K-12 education system. Although turtle programming is now used more as a tool to foster computational thinking, turtle geometry has also been explored as a powerful way of teaching geometry and mathematical reasoning to children (Hoyles & Noss, 1992; Clements & Sarama, 1997). In turtle geometry, a turtle acts as a programmable object that navigates the screen, drawing as it goes and turning at specified angles, to create simple visual patterns. The primary objective within this framework is to generate code capable of producing simple visual inputs. These visual inputs consist of basic geometric shapes, and the programming syntax required is intentionally limited and straightforward. An example of such a task is presented in the left side of Figure 1. As illustrated, the input image is the shape of a simple square and the corresponding code only uses two simple turtle functions (`forward` and `right`) along with a simple for loop. This simplicity makes TurtleBench an effective benchmark for evaluating the capabilities of LMMs.

To reflect different real-world use cases of an LMM in the domain of Turtle and also cover the broad range of underlying reasoning abilities, TurtleBench includes 260 tasks with a variety of types and modalities. We conduct an evaluation of leading LMMs on TurtleBench code generation and code editing tasks, utilizing zero-shot and visual chain-of-thought (Singh et al., 2023) approaches across text-only, image-only, and mixed (text and image) input modalities. Our findings reveal that these models generally perform poorly across all setups and variety of tasks and modalities. Our best-performing model, GPT-4V, outperforms Gemini 1.5 Flash yet neither model comes close to solving TurtleBench tasks, as about 75% of the tasks were left completely unsolved. Intriguingly, our results indicate that performance improves when tasks are presented in text, rather than inputting images. This suggests that integrating visual and linguistic information, particularly in domains requiring visual pattern recognition, may need further refinement. All these findings demonstrate that our benchmark poses a challenging task for LMMs, providing valuable insights into their capabilities.

## 2 🐢 Overview of TurtleBench

TurtleBench is a set of 260 tasks that are designed to evaluate LMMs' performance on vision and language algorithmic reasoning tasks. To ensure the novelty of the tasks and their quality in incorporating authentic geometric shapes and concepts, we craft TurtleBench manually. All the tasks

in TurtleBench are accurately solvable based on the provided information for each, which means that there are no ambiguities or arbitrary parameters leading to inaccuracies in the tasks for humans as well as the models. To remove possible ambiguities in the tasks, two independent annotators worked with us to identify and resolve any unclear instructions. Each task consists of a black-and-white image illustrating a set of abstract geometric shapes as an *input*. An example of this task is presented in Figure 1. TurtleBench is made up of two different types of tasks, these types reflect the methodologies used in turtle geometry to introduce programming to children.

*Scratch* tasks are intended to show how well a model understands a pattern and translates its understanding to an executable code. In the general case of this type of task, an image is provided, and the requested output is code in Python Turtle that creates the shapes in the image. In all scratch tasks, the model is asked to *generate* the code in Python Turtle for the desired input shape. TurtleBench includes a total of 130 scratch tasks. An example of these tasks is provided in Figure 1, top rows. To distinguish between the models' visual comprehension and their textual understanding, a subset (31%) of these tasks includes a text description of the image input in addition to the visual representation. This setup facilitates the evaluation of how models respond differently to visual and textual inputs, providing a clearer understanding of their capabilities.

*Tweak* tasks are intended to measure how well a model uses their understanding of a visual pattern, combined with an instruction to make minimal alterations. Each tweak task presents a model with an image and an instruction; the expected output is Python Turtle code that modifies the shape in the input image according to the given instruction. These tasks are particularly insightful for determining whether a model is merely recalling memorized code for an image, or if it has developed a deeper, more human-like comprehension of the patterns depicted in the images. For instance, a model might be capable of generating code for a certain shape based on training data, but the real challenge lies in its ability to adapt that shape in response to various instructed changes. An example of these tasks is provided in Figure 1, bottom row. Here, the model is given an input image of a rectangle, with an instruction to *connect the midpoint of each side to the midpoint of adjacent sides*. As illustrated in Figure 1, we also introduce a code editing version of the tweak task. In this version, we supply the code corresponding to the input image and then instruct the models to make specific modifications to this code, aiming to achieve a change in the image as per the provided instructions. Detailed information about types of tweaks and their examples is provided in Appendix C.4.

## 3 Evaluation Setup

In the following section, we evaluate TurtleBench using two state-of-the-art LMMs, GPT-4V and Gemini 1.5 Flash and also an open source model, namely Llava-1.5-13B(Liu et al., 2023) employing greedy decoding in our evaluations. We evaluated two other open models, namely Qwen-VL-Max (Bai et al., 2023) and CogVLM (Wang et al., 2023) on a subset of tasks in TurtleBench. However, CogVLM and Qwen are not successful in producing a syntactically correct Python Turtle piece of code even for the simplest tasks, therefore we limited our benchmark evaluation to the models mentioned above.

We utilize two types of prompting in our experiments, 1) basic, where we simply prompt the the model (c.f. Appendix C.2) to do our tasks, and 2) Chain-of-Thought (CoT) prompting (Wei et al., 2022), which has shown to be an effective prompting technique in eliciting reasoning in these models. Specifically, we use a more detailed version of CoT prompting that is tailored to LMMs, namely v-CoT, recently proposed by Singh et al. (2023). The v-CoT approach is inspired by m-CoT (Zhang et al., 2023), which shows higher performance compared to it. This prompting has been shown to improve LMMs' performance on visual tasks that involved reasoning, such as ARC (Chollet, 2019). This prompt, instructs the model to first extract all the relevant information in the image needed for answering the problem and then to reason step by step based on the information extracted. The specific prompt we used in our experiments is in Appendix C.2

## 4 Results

### 4.1 Models perform poorly on TurtleBench

We initially examine the performance of the GPT-4V, Gemini 1.5 Flash and Llava-1.5-13B models on the comprehensive TurtleBench dataset. The findings, detailed in Table 1, reveal a notably poor

| | GPT-4V basic | Gemini basic | GPT-4V 0-S CoT | Gemini 0-S CoT | Llava-1.5 basic | Llava-1.5 0-s CoT |
|---|---|---|---|---|---|---|
| *Scratch Code Generation* | | | | | | |
| Image only | 16% | 7.7% | 19.23% | 8.46% | 1% | 1% |
| *Tweak Code Generation* | | | | | | |
| Image + Text | 10% | 3.85% | 12.3% | 7.7% | 0% | 1% |
| *Tweak Code Edit* | | | | | | |
| Image + Text | 18% | 12% | 18.46% | 18.46% | 1% | 1% |
| Image + Image | 12% | 3% | 13.84% | 8.46% | NA | NA |

Table 1: Performance of GPT-4V, Gemini 1.5 Flash, and Llava-1.5-13B on TurtleBench. Our result shows that models perform poorly on TurtleBench.

performance across the tasks in TurtleBench, with a peak accuracy of 20% achieved by GPT-4V in the *code editing* tasks, facilitated by Chain of Thought (CoT) prompting. In the *scratch* tasks, which represent the simplest problem type within the dataset, GPT-4V's success rate was just 19%, underscoring the substantial challenges and complexities these tasks pose to the current models. A comparison between CoT and basic prompting within Table 1 illustrates that CoT prompting outperforms basic prompting on the same models, aligning with previous work that indicates CoT enhances models' reasoning abilities (Zhang et al., 2023). However, despite utilizing CoT prompting, the task remains far from being solved. Additionally, we note a decline in the performance of models when comparing tasks that involve tweaks to those starting from scratch. This observation suggests that models fail to generalize their understanding to tweak tasks, even if they can successfully complete tasks from scratch. Examples of model output in different subsets of the task are provided in Figures 8 and 10.

### 4.2 Limited Visual Understanding in LMMs: Insights from Textual vs. Visual Tweak Tasks

For tweak tasks, where the AI had to edit existing code, we gave instructions either in natural language or as images (see Figure 1, bottom rows, left two columns). As can be seen by comparing the bottom two rows in Table 1, there is a huge decline in accuracy when instructions were provided visually rather than textually, especially for Gemini. This outcome suggests a disparity in the models' ability to process visual versus textual instructions, revealing that their reasoning abilities may not align closely with human-like understanding. The assumption that directly viewing the desired outcome simplifies the task contrasts sharply with our findings, highlighting a reliance on textual interpretation for reasoning and a notable limitation in pure visual reasoning capabilities within these models. In Appendix B.3, we provide further evidence of this with additional analyses on scratch tasks by varying the input to those tasks (i.e., visual or textual descriptions).

## 5  Conclusions

This study introduces TurtleBench, the first of its kind in benchmarks that focus on converting visual inputs to code outputs. The evaluation results from TurtleBench reveal a significant disparity between human capabilities and current state-of-the-art AI models in understanding simple geometric shapes, reasoning about these shapes, and converting such understandings into executable code. This gap underscores the challenges that lie ahead in the quest to enhance AI's comprehension and problem-solving abilities to match human levels. We believe that TurtleBench serves as a crucial tool in the evaluation of models, offering a clear benchmark that tests the limits of large multimodal models.

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

## A  Related Work

### A.1  Large Multi-modal Models

Recent advancements in foundational multimodal models have marked a significant stride towards developing generalist AI systems capable of understanding and integrating information across different modalities to solve tasks without the need for task-specific fine-tuning. Among these models are closed source models such as Gemini 1.5 Flash (Team et al., 2023), GPT-4V (OpenAI et al., 2024), and open source models as LLaVA-1.5 (Liu et al., 2023), Mini-GPT4 (Zhu et al., 2023), InstructBLIP (Dai et al., 2023) and CogVLM (Wang et al., 2024). The versatility and multimodal understanding exhibited by these foundational multimodal models have positioned them as prime candidates for applications such as AI software engineers or programming tutors for children. Our work evaluates the efficacy of these popular models on image/text-to-code tasks, measuring their potential in vision/programming context.

### A.2  Probabilistic Program Induction

Recent work in Bayesian cognitive science has modeled various aspects of cognition and learning as probabilistic program induction (Lake et al., 2015; Lake & Piantadosi, 2020; Rule et al., 2020; Ellis et al., 2023; Wong et al., 2021; Grand et al., 2023). This has involved both modeling human cognition as program induction as well as designing machine learning algorithms that can generate programs for various tasks, including the kind of turtle geometry task we study here. Ellis et al. (2023) developed the DreamCoder algorithm which can learn to induce programs by using self-supervision to incrementally build up a library of programs and train a neural network to search to find the best program for a given task. They created a dataset of 160 turtle programming tasks. In contrast to our approach, where we assess the performance of out-of-the-box LMMs, DreamCoder is trained on a training set of images (i.e., half of the dataset). However, it is interesting that the algorithm is trained in an unsupervised fashion; that is, DreamCoder never receives the code used to generate the images and learns that from experience. Wong et al. (2021) extended this work by developing an algorithm (LAPS) that can induce programs given both the task and linguistic annotations for the task. They

used a dataset of 311 turtle graphics with greater complexity than the original DreamCoder dataset. While their dataset includes linguistic annotations, their dataset does not include tweak tasks like in TurtleBench. Additionally, their tasks often include arbitrary aspects (for example, a gap with unspecified distance between two shapes) that makes evaluation hard; in our tasks, the positional relationships between shapes should be easy to infer exactly and hence we can evaluate models by comparing exactly with ground truth shapes. Moreover, neither of these datasets have been framed as a benchmark for visual program induction and have not been considered for evaluating LMMs. Perhaps the approach closest to our work is by Grand et al. (2023), who combined LLMs with a symbolic program induction algorithm and evaluated the performance of their model (LILO) on the turtle geometry task using the aforementioned dataset. Averaged over several runs, the performance of the best versions of these approaches on the turtle geometry task is as follows: 43% for DreamCoder, 82% for LAPS, 49% for LILO, and 32% for a LLM solver. These results seem to suggest that probabilistic programming approaches (such as LAPS) can greatly outperform LMMs on visual programming tasks. We note that the performance of the LLM solver (32%) is comparable to the performance of GPT-4V on our text-only input (37%; see Table 4). Future work could assess the performance of probabilistic program induction methods like LAPS on TurtleBench.

### A.3 Mutimodal Algorithmic Reasoning

The existing literature features a range of studies that evaluate these models using naturalistic images (Jiang et al., 2022; Johnson et al., 2017; Antol et al., 2015), yet humans naturally are able to reason over abstract shapes (Chollet, 2019; Zhang et al., 2019; Spelke & Kinzler, 2007) and also many use cases of LMMs involve understanding abstract shapes and sketches (Forbus et al., 2011; Nie et al., 2020). Moreover, unlike naturalistic images (Marjieh et al., 2022; Sucholutsky & Griffiths, 2024), the relationship between language and abstract shapes is highly intertwined as minimal alterations in language can lead to different visual perceptions in humans (Dillon, 2023; Lin & Dillon, 2023). The Multimodal Algorithmic Reasoning (MAR) task tests multi-modal models on fundamental skills understandable by children, focusing on interpreting visual and linguistic information to answer questions. Perhaps the most relevant work to ours is the paper by Cherian et al. (2023) in which they introduced a dataset with 101 multiple-choice questions inspired by the Math Kangaroo contest for 6 to 8-year-olds, involving images and texts that the model must analyze together. The task has been shown to be challenging for multimodal deep neural networks, and the following trials to solve the problem have gained less than 25% accuracy on the private test set Wu et al. (2023). Our proposed benchmark pushes the evaluation of LMMs forward as TurtleBench includes abstract geometric shapes, and the task only relies on knowledge and reasoning over a set of simple functions in the Python Turtle library. The open-ended nature of our benchmark and its flexibility over different modalities makes evaluating different aspects of vision and language algorithmic reasoning in the models more reliable.

## B  Additional Analyses

### B.1 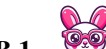 Models fail to generalize

Given that these models have been extensively trained on vast datasets sourced from the internet, there's an underlying uncertainty regarding the source of their performance—albeit poor—on the TurtleBench tasks. Specifically, it remains unclear whether this performance is the result of the models' ability to memorize aspects of our tasks, rather than genuinely understanding and solving them based on their programming and reasoning capabilities. To address this issue, our next step is to evaluate the true generalization ability of these models. By doing so, we aim to distinguish between superficial learning, potentially influenced by memorization, and genuine comprehension and problem-solving skills. To measure the generalizability of the model's performance, we define an arbitrary set of commands based on the turtle module in Python. In other words, we developed a class called Rabbit that inherits the `Turtle` class from the turtle module. Although the functions of the Rabbit class are functionally identical to those in the original turtle module, they are nominally distinct. This differentiation allows us to evaluate the models' ability to apply their knowledge to unfamiliar yet equivalent command sets. The definition of the Rabbit class in Python is provided in Appendix C.3.2. We perform a zero-shot CoT prompting to elicit the code using the new set of

|  | 🐢 GPT-4V Turtle CoT | 🐰 GPT-4V Rabbit CoT | 🐢 Gemini Turtle CoT | 🐰 Gemini Rabbit CoT |
|---|---|---|---|---|
| *Scratch Code Generation* Image only Input | 19% | 6% | 8.46% | 3% |
| *Tweak Code Generation* Image + Text | 12% | 2% | 7.7% | 1% |

Table 2: Performance of GPT-4V and Gemini 1.5 Flash on generalization tasks, in these tasks, we defined Rabbit, a new set of functions practically equivalent to but nominally different from the ones in Python Turtle. The performance in Rabbit drastically drops, showing poor generalization abilities in both models.

|  | 🐢 Python Turtle Output | Any Output |
|---|---|---|
| *Scratch Code Generation* Image only Input | 19.23% | 21.6% |
| *Tweak Code Generation* Image + Text | 12.3% | 15.1% |

Table 3: Performance of (CoT) prompting with GPT-4V on tasks involving code generation for simple geometric shapes in any programming language of the model's choice reveals that models struggle significantly, even in their preferred programming language.

commands. In the context window, we provide a verbal definition of each function in the Rabbit class. The results of comparing the models' performance using the Rabbit class versus the standard Python Turtle module are presented in Table 2. We observe that, although both models were capable of generating executable pieces of code with the new class, there is a huge decline in their performance relative to their performance with the conventional Python Turtle module. This finding suggests that the visual reasoning in these models is not robust to syntax changes, and it is likely that they rely on training memorization rather than pure reasoning.

## B.2    Assessing Model Proficiency Across Programming Languages

The initial suspicion might be that the models struggle with tasks in turtle geometry due to a lack of exposure to specific programming syntax during pretraining. However, to investigate whether the challenge lies not in syntax familiarity but in understanding visual input and translating this understanding into effective programming, we modify our approach with GPT-4V. We choose GPT-4V as it is our best-performing model in the main task. We allow it to generate code using any library, language, or similar tools it deems appropriate, such as Matplotlib, TikZ, etc., without restricting it to the Python Turtle library. The prompt for this subset of tasks is presented in Appendix C.2.4. We manually evaluate the GPT-4V output for this task. Despite this freedom, we observe no significant improvement in performance. The model chooses Matplotlib for 50% of the tasks and offers pseudocode for 2%, with the remainder reverting to Python Turtle, even though we do not specify Python Turtle in the prompts. Notably, it avoids using TikZ, despite its mention in the prompt and proven capabilities in prior work to produce TikZ code (Bubeck et al., 2023; Belouadi et al., 2023). This outcome underscores a deeper issue than syntax familiarity: the models' fundamental challenge is accurately interpreting visual input and applying this understanding to generate corresponding programming code.

## B.3    Limited Visual Understanding in LMMs: Insights from Scratch Tasks

One of the questions regarding LMMs' abilities in visual abstraction and understanding tasks is the extent the incorporation of the visual component has enhanced their abilities in reasoning (Mitchell et al., 2023). In resonance with what Mitchell et al. (2023) found, here we also found that the vision component contributes poorly to fostering the models' visual reasoning abilities, at least in the domain of TurtleBench. We explored this in the context of tweak tasks in Section 4.2. Here, we explore it in

|  | GPT-4V basic | Gemini basic | GPT-4V CoT | Gemini CoT |
|---|---|---|---|---|
| *Scratch Code Generation* | | | | |
| Image only Input | 26% | 7.7% | 29% | 8.46% |
| Text only Input | 37% | 25.1% | 38% | 18.51% |
| Image and Text Input | 38% | 22.2% | 40% | 22.22% |

Table 4: Performance of GPT-4V and Gemini 1.5 Flash on TurtleBench, for comparing visual vs. text input on Scratch Code Generation Tasks

the context of scratch tasks. Specifically, we annotated 41 scratch code generation tasks and provided clear descriptions for each in plain text. The remaining shapes were too complex to describe without ambiguity in plain text. Then, we compared the three modes of presenting the task, image only, text only, and the blend of an image and its description in text. Interestingly, for both GPT-4V and Gemini 1.5 Flash, the model performed worse when the task was presented only in the image, compared to the other modes. This phenomenon is counterintuitive as for humans, perceiving the images should be easier than first reading a description, imagining it, and then writing a code for it. Additionally, as presented in Table 4 the blend of image and text only slightly improved GPT-4V's performance (from 38% to 40%). These two findings show that there is still much room for improvement especially in the visual components of LMMs.

## B.4 Reasons of Failure

We manually investigated GPT-4V's failures in solving Scratch tasks in a single run to find the major causes of failure. We find four major causes: 1) Shape identification error: where the model fails to completely capture existent shapes in the input image, for instance, if it confuses a semicircle with a circle or assigns non-existent shape attributes to the input image. 2) Counting error: where the model fails to count adequately, (e.g., three triangles counted as four), 3) Orientation error: where the model fails to correctly find the relationships between different components of a shape (e.g., semicircle on top of a square vs. at its bottom), and 4) Implementation error: where the model's generated code does not follow the pre-planned pseudocode.

We manually investigated GPT-4V's failure output in the scratch code generation task and the results are provided in Table 5, where the failures are not mutually exclusive as a model can perform a combination of errors in each task. Furthermore, while the first three errors are according to the vision component in these models, we see that 64% of the failures are according to these causes, and in 36% of failure cases, there are no apparent vision errors.

| Cause | Description | Percentage |
|---|---|---|
| Shape identification error | **Shape Identification Error:** The model fails to completely capture existent shapes in the input image, confusing or misattributing shapes | 25% |
| Counting error | **Counting Error:** The model inadequately counts the elements. | 35% |
| Orientation error | **Orientation Error:** The model fails to correctly determine the spatial relationships between different components of a shape | 21% |
| Implementation error | **Implementation Error:** The model's generated code does not adhere to the pre-planned pseudocode, resulting in incorrect implementation. | 45% |

Table 5: Major Causes of GPT-4V's Failures in Scratch Tasks; note that the failures are not mutually exclusive, as a model can perform a combination of errors in each task

# C  Experiment Setup

## C.1  Automatic Evaluation of Code Output

Evaluation of the output code by an AI model is performed automatically. First, the output of the AI model is processed to extract the code piece of output. Then, this piece of code is run in a sandbox, and the shape produced by the code is stored. An illustration of this pipeline is provided in Figure 9. Finally, using the *OpenCV* module in Python, the binary versions of the correct shape and the produced shape are compared using an adjusted measure of bitwise similarity where we first use the bounding box technique with *OpenCV* to find the exact location of the shape and then calculate similarity with the formula:

$$\frac{|B_a \cap B_m|}{|B_a \cup B_m|}$$

where $B_a$ and $B_m$ represent black pixels in the input and LMM output, respectively. This metric measures the ratio of co-occurring black pixels to the total black pixels Here, we utilize a heuristic approach in labeling the correctness of the model's output. If the bitwise similarity between output and ground truth is higher than 95% the models' output is labeled as correct and incorrect otherwise. To make sure that our heuristic in labeling the correctness of generated shapes is reliable, we manually annotated 2000 pairs of input and output images and we found that only three instances of pairs were labeled incorrectly (two of them false negative and the other false positive.), leading to an error rate of 0.15% which shows the high level of reliability in the heuristic we used.

## C.2  Prompting

### C.2.1  Basic Prompt

```
In each task, the user provides an image of an abstract geometric shape or pattern
and an instruction, you need to generate a code in Python Turtle that follows the
user's request.
```

Figure 2: basic prompt used in our experiments

### C.2.2  v-CoT Prompt

```
You are Turtle Geometrician, you are an expert in reasoning about images and
generating code in Python Turtle using images You need to follow the steps below
before generating the answer:
(1) Describe the relevant information from the image needed to answer the question.
List all relevant artifacts from the image.
(2) Use the information described in (1) to reason about the problem by working
step by step to arrive at the final piece of code.
(3) Generate the final code. NEVER use "pensize" function in your code.
```

Figure 3: v-CoT prompt used in our experiments

### C.2.3  A Complete Example

Here we provide an instance of a complete prompt we used for a tweak *code generation* task with CoT prompting:

### C.2.4  Arbitrary Output

Here we provide the CoT prompt we used for the model to provide a code in any arbitrary language or library that creates the desired shape.

Figure 4: An example of a complete prompt for a tweak code generation task with using v-CoT prompting.

```
You are an expert in reasoning about images and generating code in any language you
prefer. You need to follow the steps below before generating the code that answers
the user's request:
(1) Describe the relevant information from the image needed to answer the question.
List all relevant information from the image.
(2) Use the information described in (1) to reason about the problem by working
step by step to arrive at the final piece of code.
(3) Generate the final code. Your code can be in any visual language or library,
such as Matplotlib, TikZ, etc.
```

Figure 5: The system prompt we used for the results discussed in Section B.2

## C.3    Rabbit

### C.3.1    Prompt used

The prompt we used for this experiment is provided in Figure 6.

### C.3.2    Definition of the class

The rabbit class is an arbitrary class that we defined based on *Turtle* class in the Python Turtle Module. This minimal set of functions includes all functions that a programmer or a model needs to create all of the tasks in TurtleBench. We defined this new set of functions to measure how GPT-4V is able to generalize its abilities in generating code in Python Turtle to a similar but minimally different set of functions.

```python
import turtle

class Rabbit(turtle.Turtle):
    def __init__(self):
        super().__init__()
        self.setheading(90)
        self.pensize(5)
        self.hideturtle()

    def aa(self, length):
```

```
Suppose that I have a library named Rabbit in Python. Rabbit library has an object
constructor named Rabbit which is an object that moves on the screen and draws
lines. It only has these functions:
aa(length): goes front or back (if the length is negative) and draws a line with
the length of pixels.
bb(degree): The rabbit turns its head right or left (if degree is negative).
cc(radius, degree): creates an arc with the given radius for the given degree. If
degree=360 it creates a circle. The center of the circle is in the left of the
rabbit.
pp(vanish): if vanish=True vanishes Rabbit object so if it moves does not draw
anything, and if vanish=False, it appears the Rabbit object so if it moves draws on
the screen.
you call the functions on an object of Rabbit, such as r.aa(length) where r is an
object of Rabbit. When r is created, it faces north (up) on the screen and it does
not vanish, so it is in drawing mode.

You are Rabbit Geometrician, you are an expert in reasoning about images and
generating code in Python Rabbit using images. You need to follow the steps below
before generating the answer:
(1) Describe the relevant information from the image needed to answer the question.
List all relevant artifacts from the image.
(2) Use the information described in (1) to reason about the problem by working
step by step to arrive at the final piece of code.
(3) Generate the final code. Only use commands in the Rabbit class.
```

Figure 6: v-CoT prompt used for generalization experiments discussed in Section B.1

```python
        self.forward(length)

    def bb(self, degree):
        self.right(degree)

    def cc(self, radius, degree):
        self.circle(radius, degree)

    def pp(self, vanish):
        if vanish:
            self.penup()
        else:
            self.pendown()
```

## C.4 Types of Tweak Tasks

TurtleBench includes a total of 130 tweak tasks. We provide a categorization for the tweaks as follows: There are five major types of tweaks in TurtleBench;

- Deletion: Removing a specified part of a shape
- Insertion: Adding a specific shape to the pattern as directed
- Rotation: Rotating the entire shape
- Reflection: Reflecting the entire shape or parts of it across specified lines
- Generalization: maintaining a pattern in the image constant while varying its parameters.

An illustration of instances of each type is provided in Figure 7. These types are not mutually exclusive as 10% of the tasks involve a combination of two types (e.g., removing one side of a square and inserting a semicircle instead). To successfully complete deletion and insertion tweaks, a model needs to demonstrate a nuanced understanding of the details in the image and program the resulting shape accordingly. In contrast, rotation tasks can be relatively easy as most of them can be solved only using a simple function in Turtle that can rotate the starting heading of the turtle which results in complete rotation in the entire shape (i.e., turtle.right(angle)).

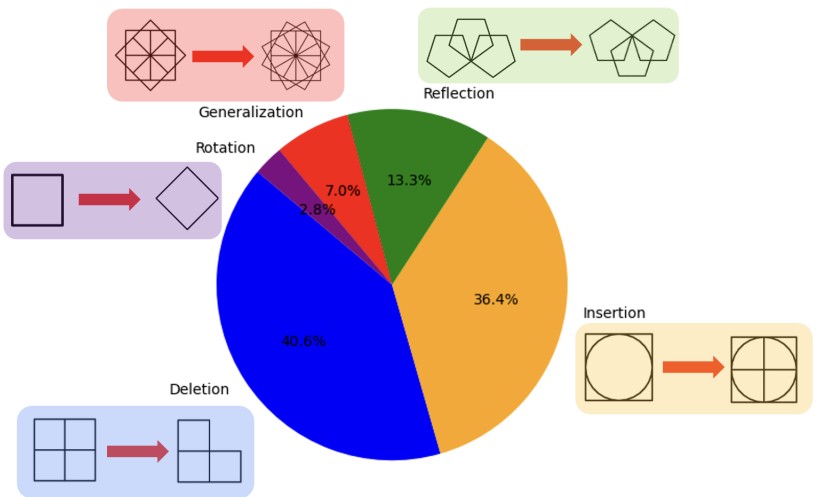

Figure 7: Types of tweaks and their share in TurtleBench

## C.5 Evaluating Image Complexity Using Contour Counts

As our result suggests that the vision component is contributing poorly to the models' performance, to gain a better understanding of the visual obstacles for the models to solve the tasks, we defined a measure as a proxy for the complexity of shapes. For each provided image, we calculated the number of contours in each shape. In OpenCV, a contour is a curve joining all the continuous points (along the boundary), having the same color or intensity. Contours are a useful tool for shape analysis and object detection and recognition. The high number of contours in an image hints that there are many shapes being involved and interleaving with each other, which makes understanding and extracting underlying patterns challenging.

We calculated the number of contours in each shape by utilizing the corresponding function in OpenCV, and defined three arbitrary levels of complexity in the images, where the images which include only one contour (e.g., the basic square in Figure 1) are at level 1 (simple), images including less than 6 contours and more than 1 are at level 2 (medium) (e.g, the base shape of insertion example in Figure 7) and the images in which there are more than 6 contours (e.g., the base shape in generalization example in Figure 7) are at level 3 of the complexity (Complex). In Turtle, the proportions of complexity levels 1, 2, and 3 are 25%, 40%, and 35%, respectively.

We investigate how models perform over tweak tasks. There are 9 different ways that a pair of input and output image can combine. As shown in Table 6, the majority of tweak tasks (74) have same levels of complexity for the input and output image.

To examine how complexity of input and output shapes impact the results, we categorize tweak tasks in the 9 different categories and count the number of tasks that are ever solved by GPT-4V under any prompting method in code generation and code edit tasks during 6 different runs. As shown in Table 6, the more complex the input shape is, the more challenging solving the task is.

|  |  | Output Complexity | | |
|---|---|---|---|---|
|  |  | Simple | Medium | Complex |
|  | Simple | 35% (7/20) | 30% (3/10) | 25% (1/4) |
| Input Complexity | Medium | 40% (2/5) | 18% (6/33) | 7% (1/13) |
|  | Complex | 20% (1/5) | 11% (2/19) | 19% (4/21) |

Table 6: The number of tweak tasks under each category and the percentage of those tasks ever solved by GPT-4V in different settings.

## C.6 Task Instances

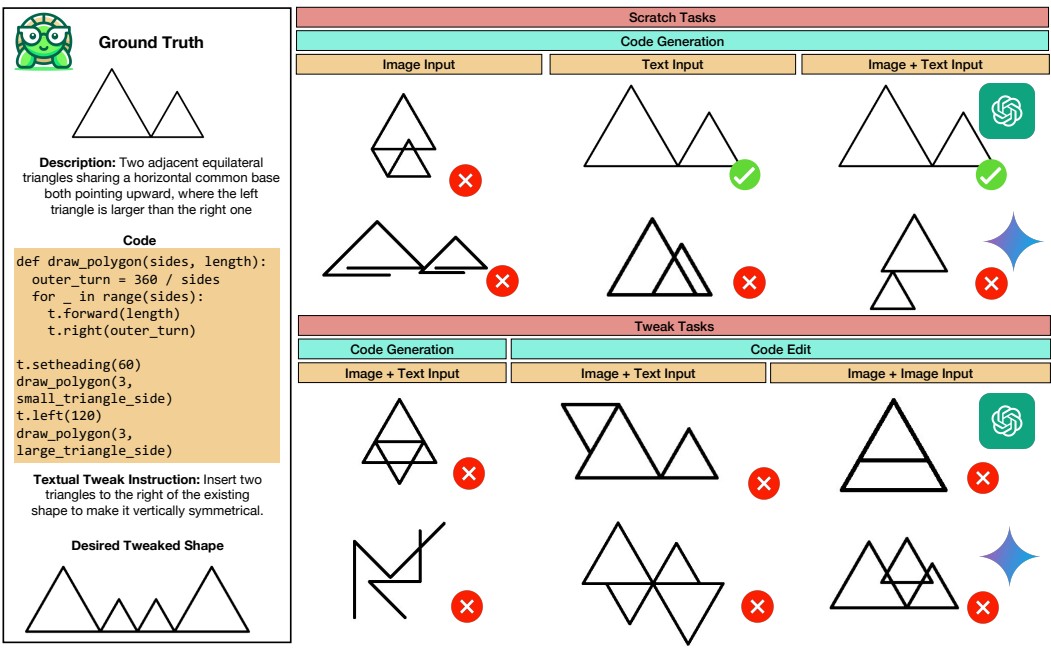

Figure 8: An illustration of different modes of a single task in TurtleBench along with the images generated by code from the outputs of GPT-4V and Gemini 1.5 Flash. More examples are provided in Appendix Figure 10

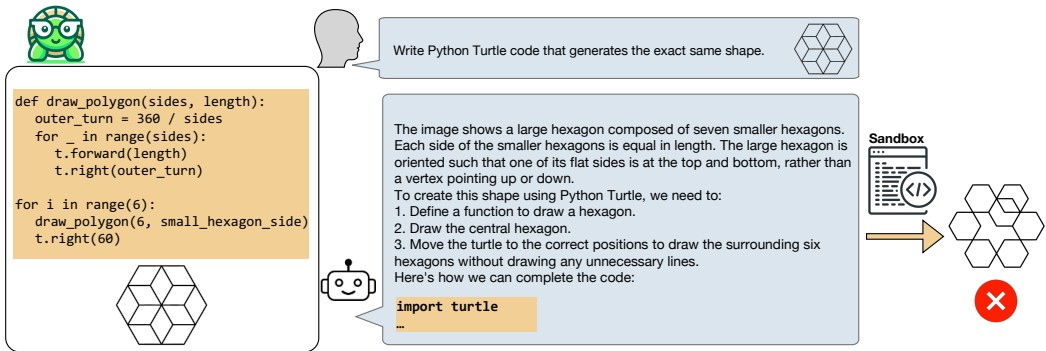

Figure 9: An illustration of our evaluation pipeline

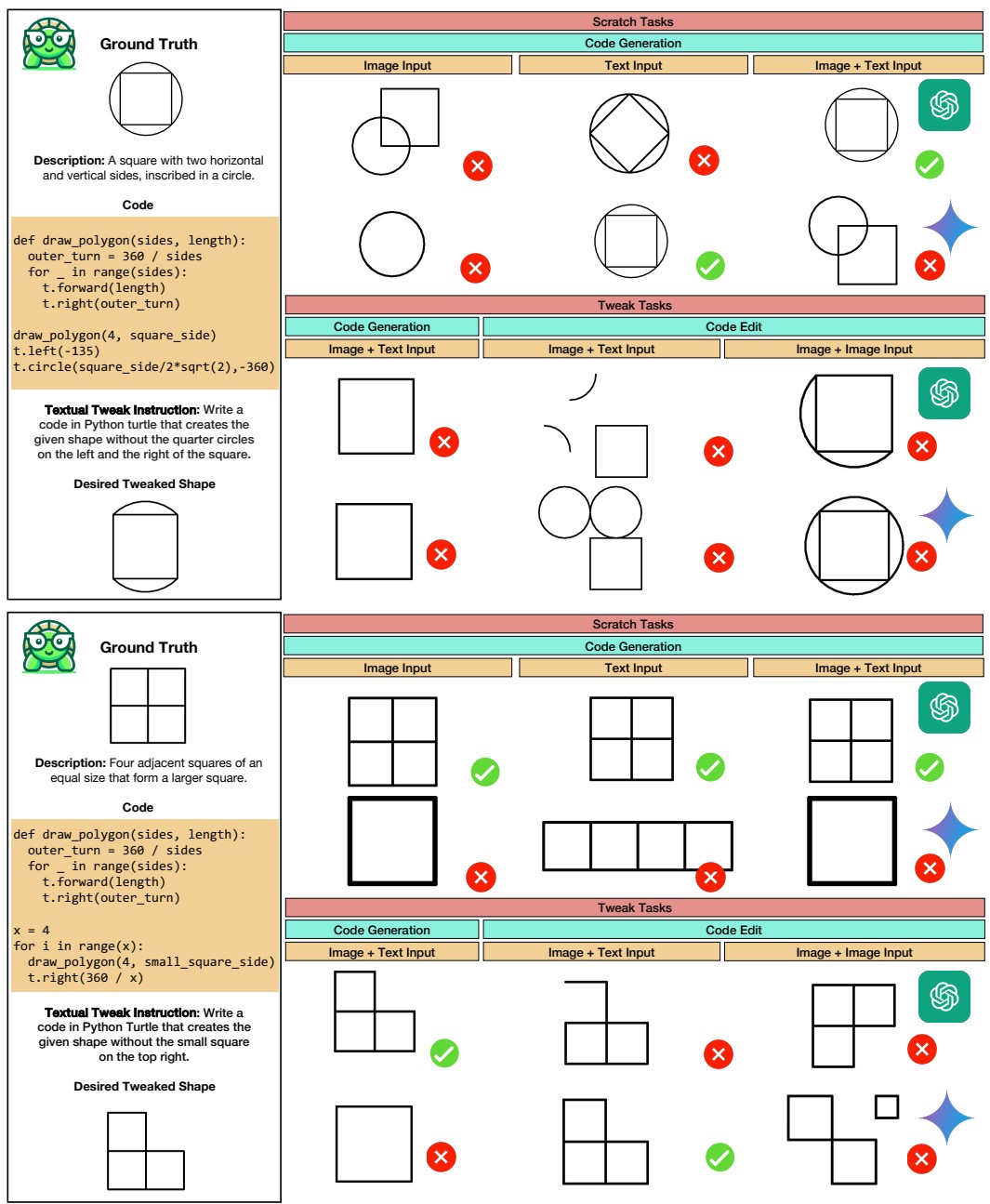

Figure 10: Two examples of tasks in TurtleBench across different modalities

