# OpenReview forum: "TurtleBench: A Visual Programming Benchmark in Turtle Geometry"
_NeurIPS.cc/2024/Workshop/MATH-AI — MATH-AI 24_

### Official Review · Reviewer_FVHh · 2024-09-29
**# Review of Paper 48: TurtleBench: A Visual Programming Benchmark in Turtle Geometry**

**Rating:** 6
**Confidence:** 3

**Review:**

# Review of Paper 48: TurtleBench: A Visual Programming Benchmark in Turtle Geometry

# Reviewer Summary of paper:

The paper focuses on geometry problems using the Turtle programming language.
The consequence is a novel variation on multimodal data in the Math AI space, text + code + image as input.
The authors evaluate off the shelf models include closed models like gpt-4v and gemini as well as open source ones like LLava.
Some models were excluded from preliminary checks which indicated extremely poor performance.
The evaluation focused on two types of tasks (i) `scratch` and (ii) `tweak`. The `scratch` tasks require *de novo* code generation whereas the tweak tasks require code editing.
In general the authors note poor performance on the tasks, performance is slightly enhanced by visual-COT prompting.

# Major points:

1. lines 65-67, "This suggests that integrating visual and linguistic..., may need further refinement". This statement aligns with the generally known need to train or fine tune domain specific models.
For the purely text input case cf. "Training Verifiers to Solve Math Word Problems" 2021 openai tech report, figure 3 which demonstrates improvement (test@1). For the image + text (multimodal), cf. "Smart Vision-Language Reasoners" ICML 2024, which show general improvements for image + text in aggregate (similar to prior findings on text only) as well as in individual problem classes (geometry, algebra, counting, etc). FunSearch (google deepmind Dec 2023) has investigated text and code inputs but not vision.

Suggest that this line adds citations to these three manuscripts as prior work.

2. For how many tasks were the evaluations in Section 3 conducted?

Appendix c.1 describes the particulars of the code output evaluation which was conducted on 2000 tasks, however, no task count is given for any other evaluations conducted in Section 3. In Section 2 where the `scratch` and `tweak` tasks are defined, the author(s) mention 130 scratch tasks but no mention of the number of `tweak` tasks.

Could the author(s) please clarify in revision?

3. I was not able to comprehend the distinction between a general `tweak` task and an `edit` version of the `tweak` task as defined in line 98. Suggest to use either a single descriptor word e.g. pick either `tweak` or `edit`, and use the single word throughout the manuscript, or add a description of the distinction between the two and the relevance for the distinction.

# Minor points:

1. For the visual instructions in Section 4.2 it is not clearly described. Did the author(s) generate a visual depiction of the geometric construction (e.g. visually depicting the geometric constructions) or did they generate an image of the textual description?

2. The references section contains multiple references, e.g. lines 433-438

line 124 change the 20% to 19.23% to conform with the associated entry in the table, doing so aids the reader is identifying supporting evidence of the claim and placing the claim in context.

2. In both the `scratch` and `tweak` tasks, how was success defined?

Some approaches generate multiple solutions (up to 400 have been reported-that I know of) and then rank them and pick the best (cf. "Training Verifiers to Solve Math Word Problems", section 5 intro paragraph). Others may use a single generated solution or somewhere between the two extremes. No mention of a verifier approach was mentioned here so I suspect only a single solution generation was conducted. Stating the difference here aids the reader in placing the results in context with other papers performance claims.

# Some optional/suggested further ablations:

1. In the ablation study of the visualization library, the authors note that 50% of the time the output chose `matplotlib`. If the authors modify the prompt to specifically ask for matplotlib, perhaps performance would be improved? Assessing the import of the choice of visualization library is an important bit of information which warrants a deeper look.

2. The tweak tasks are interesting assessments of code and problem processing which are more challenging for evaluated models to perform. A likely further step in assessment would be to use a tool approach and have the tool be a (domain specific) code model.

---

### Official Review · Reviewer_s5Z6 · 2024-10-07
**Good benchmark for visual programming and reasoning in terms of code generation and code edit.**

**Rating:** 7
**Confidence:** 4

**Review:**

This paper introduces TurtleBench, a benchmark designed to evaluate large multimodal models (LMMs) on their ability to interpret geometric patterns through visual examples and textual instructions. The study highlights the significant challenge for LMMs in this domain, as even state-of-the-art models like GPT-4V perform poorly, emphasizing the gap between human intuitive geometrical reasoning and current AI capabilities.

Pros:
1. TurtleBench will benefit a lot in this field, with scratch and tweak tasks in terms of code generation and code edit.

2. Detailed comparisons between various mLLMs are given in the experments, which strongly support the conclusions that there exists  vary large gap between human and AI performance in intuitive and visual geometrical understanding.

Cons:
1. What is the potential direction for visual programming instead of LLM or mLLM, e.g., on TurtleBench? It will be better if the authour can give more suggestions on this task.

2. Any other geometric reasoning or spatial reasoning benchmark for visual programming in the field of computer vision? More comparisons and discussions with these benchmarks should be added.

---

### Official Review · Reviewer_hPBk · 2024-10-09
**Interesting benchmark for future visual understanding and reasoning**

**Rating:** 7
**Confidence:** 3

**Review:**

This paper introduces TurtleBench, a benchmark designed to evaluate the ability of large multi-modal models (LMMs) to interpret geometric patterns.
The benchmark includes two tasks: Scratch and Tweak.
Models such as GPT-4V Basic, Gemini, and Llava have demonstrated poor performance on these tasks.
These tasks are crucial for a comprehensive assessment of the models' multi-modal reasoning capabilities.
I look forward to future work that further enhances this benchmark.

---

### Decision · Program_Chairs · 2024-10-08

Accept